# COVID-19 community spread and consequences for prison case rates

Katherine LeMasters[1,2]*, Shabbar Ranapurwala[1,3], Morgan Maner[2], Kathryn M. Nowotny[4], Meghan Peterson[2], Lauren Brinkley-Rubinstein[2]

**1** Department of Epidemiology, Gillings School of Global Public Health, University of North Carolina at Chapel Hill, Chapel Hill, NC, United States of America, **2** Center for Health Equity Research, Department of Social Medicine, School of Medicine, University of North Carolina at Chapel Hill, Chapel Hill, NC, United States of America, **3** Injury Prevention Research Center, University of North Carolina at Chapel Hill, Chapel Hill, NC, United States of America, **4** Department of Sociology, University of Miami, Miami, FL, United States of America

* Katherine.lemasters@unc.edu

## Abstract

### Background

COVID-19 and mass incarceration are closely intertwined with prisons having COVID-19 case rates much higher than the general population. COVID-19 has highlighted the relationship between incarceration and health, but prior work has not explored how COVID-19 spread in communities have influenced case rates in prisons. Our objective was to understand the relationship between COVID-19 case rates in the general population and prisons located in the same county.

### Methods

Using North Carolina's (NC) Department of Health and Human Services data, this analysis examines all COVID-19 tests conducted in NC from June-August 2020. Using interrupted time series analysis, we assessed the relationship between substantial community spread (50/100,000 detected in the last seven days) and active COVID-19 case rates (cases detected in the past 14 days/100,000) within prisons.

### Results

From June-August 2020, NC ordered 29,605 tests from prisons and detected 1,639 cases. The mean case rates were 215 and 427 per 100,000 in the general and incarcerated population, respectively. Once counties reached substantial COVID-19 spread, the COVID-19 prison case rate increased by 118.55 cases per 100,000 (95% CI: -3.71, 240.81).

### Conclusions

Community COVID-19 spread contributes to COVID-19 case rates in prisons. In counties with prisons, community spread should be closely monitored. Stringent measures within prisons (e.g., vaccination) and decarceration should be prioritized to prevent COVID-19 outbreaks.

**Data Availability Statement:** Data cannot be shared publicly because they were obtained via a Data Use Agreement with the North Carolina Department of Health and Human Services. The

data underlying the results presented in the study are available from the North Carolina Department of Health and Human Services for researchers who meet the criteria for access to confidential data. For non-author contact information for data requests, please send data requests to Kirsten Leloudis, Program Manager, North Carolina Department of Public Health Office of Regulatory and Legal Affairs, Kirsten.leloudis@dhhs.nc.gov. The authors of the present study did not have special access privileges in accessing the data; other researchers are able to access these data through a Data Use Agreement with the North Carolina Department of Health and Human Services.

**Funding:** LBR: The Robert Wood Johnson Foundation (78304) https://www.rwjf.org/ KL: The Sheps Center for Health Services Research (5T32HS000032-33) https://www.ahrq.gov/funding/training-grants/rsrchtng.html; The Lifespan/Brown Criminal Justice Research Training Program (2R25DA037190-06) https://www.drugabuse.gov/ The funders had no role in study design, data collection and analysis, decision to publish, or preparation of the manuscript.

**Competing interests:** The authors have declared that no competing interests exist.

## Introduction

People who are incarcerated are at increased risk for COVID-19 infection and death [1]. Many single-site cluster outbreaks of COVID-19 have occurred in prisons and jails. As of September 10, 2021 over 421,000 people who were incarcerated in state or federal prison systems have tested positive for SARS-CoV-2 infection, and at least 2,574 had died from COVID-19 [1]. The rate of infection is estimated to be 5.5 times higher among people who are incarcerated than in the general public [2]. The risk is heightened due to a confluence of several factors. For example, prisons house people who have a higher burden of chronic disease [3]. Further, the built environment of prison facilities, where people often live in close, overcrowded facilities, make common prevention strategies such as social distancing from one another and from staff members–who return to the surrounding community every day—nearly impossible [4].

The health of people who are incarcerated and that of the communities to which they return are closely intertwined. Further, COVID-19 spread in carceral facilities may reproduce and exacerbate health inequities in the general population. Black and Hispanic people are much more likely to die from COVID-19 than those who are white [5], and are also much more likely to be incarcerated due to systemic racism with one in three Black men and one in six Latino men born in 2001 going to jail or prison at some point in their lifetime as opposed to one in seventeen white men [6]. Incarceration, systemic racism, and COVID-19 therefore may operate as syndemics that mutually exacerbate one another and drive inequities [7].

Prior studies have suggested that carceral facilities (e.g., prisons, jails) can serve as points where COVID-19 infection spreads rapidly and results in spikes in community case rates. One study found that jail-community cycling in spring of 2020 predicted COVID-19 spread in Chicago, accounting for over half of the variance in case rates across Chicago zip codes and over one-third of the variance throughout Illinois [8]. However, prisons, which typically house individuals with sentences of over a year, have more stable populations than jails, which typically hold individuals awaiting trial or with sentences shorter than one year, and are often thought of as more separate from surrounding communities. However, recent research suggests that COVID-19 is highly prevalent among prison staff and has been transmitted from prison staff to incarcerated individuals in prisons, leading to outbreaks [9–12]. This is because physical distancing is limited in these settings, overcrowding is prevalent, personal protective equipment (PPE) is inconsistently provided and enforced, and staff are infrequently required to be tested [13].

Given prisons' stable populations, COVID-19 infections among staff members are likely the primary vector by which COVID-19 enters and leaves prisons. However, infection among staff members is likely a reflection of community COVID-19 spread. This study expands previous work by focusing on the broader community surrounding prisons rather than only staff members. Specifically, we assess how the rates of COVID-19 transmission in the communities surrounding prisons affect COVID-19 spread within prisons.

## Materials and methods

This study was exempt by the University of North Carolina at Chapel Hill Institutional Review Board (20–3092). We conducted a retrospective study using de-identified data from the North Carolina (NC) Department of Health and Human Services to evaluate the relationship between COVID-19 community spread and COVID-19 case rates in prisons. These data include demographics, COVID-19 test result information, the facility the test was ordered from, the individual's county of residence, and the individual's occupation. The data included information for all COVID-19 tests conducted in NC between January 1, 2020 and November 29, 2020. We restricted the dataset to tests conducted between June 1, 2020 and August 31,

2020 because a lawsuit mandated NC prisons to increase testing among their incarcerated population and staff during this time period as the state's failure to protect incarcerated individuals from COVID-19 amounted to cruel and unusual punishment [14]. The court required the state to conduct one-time universal testing and ongoing randomized testing, limit transfers, and expand criteria for early release [15]. Tests had to occur within 60 days and reports from the state's DOC confirm that this testing occurred [16, 17]. Data during this time period is thus the most accurate reflection of COVID-19 caseloads in prisons although asymptomatic spread remains likely due to one-time, rather than ongoing, universal testing being conducted.

For general population data, we restricted data to individuals with a county of residence in NC (e.g., if someone in the general population had a test ordered in NC but had a county of residence elsewhere, they were excluded) to exclude out-of-state individuals.

## Study population

We examined the impact of community COVID-19 case rates on prison case rates. To do this, we created two separate data sets: 1) all prison cases and 2) all community cases. Prison cases were classified as all positive tests in which the ordering facility was a state, private, or federal prison (including staff and those incarcerated). Community cases included all other cases ordered in a NC county with a prison among county residents. To calculate COVID-19 case rates, the denominators for the general population were extracted from the American Community Survey 2019 data [18]. Denominators for the prison population data come from The Vera Institute of Justice and a report from the NC Department of Public Safety on staff at NC state prisons [19]. Information on staff population data from Butner, NC's federal prison, was obtained from their website and information on staff population data from NC's private prisons were obtained by calling the facilities.

## Outcome

Our primary outcome was a 14-day running average COVID-19 case rate in prisons. Active case counts were calculated by summing the number of positive tests in each county's population that were in prison over the past 14 days. To calculate rates, we then divided by the sum of the number of individuals in prison in the county and the number of prison staff in the county and multiplied by 100,000. We conducted sensitivity analyses in which we removed those that were known to be staff from both the numerator and denominator.

## Exposure

We defined the exposure as the first date that the county's general population had a case rate of at least 50 per 100,000 residents in the past seven days, which is defined as substantial community spread by the Centers for Disease Control and Prevention [20]. Similar to the population in prison, we calculated case counts by summing the number of positive tests in each county's general population over the past seven days. This was then divided by the county population and multiplied by 100,000.

## Statistical analysis

We used the 14-day running average of the prison COVID-19 case rates and developed a time-series spanning up to 60 days prior-to and after the county reached "substantial community spread" (120 time points). Using these data, we conducted single-series interrupted time series analyses using an autoregressive integrated moving average model to evaluate the association between COVID-19 community spread and prison rates [21–23]. This is an appropriate and

useful method when assessing changes due to an intervention or events that occur at a clearly defined point in time (e.g., the date the community reached substantial community spread). The model can be written as:

$$\text{Outcome}_{i\ t}^{*} = \beta_0 + \beta_1{}^{*}\text{time}_t + \beta_2{}^{*}\text{intervention}_i + \beta_3{}^{*}\text{trend}_{i\ t}^{*} + e,$$

Where $\beta_0$ specifies the baseline COVID-19 active case rate in prisons at time 1 (June 1, 2020), time is a continuous variable for the entire series, which has 120 time points, and $\beta_1$ specifies the pre-interruption trend of the outcome. Intervention is a binary step function variable that represents the presence or absence of substantial community spread, and $\beta_2$ specifies the absolute change in outcome immediately when the substantial community spread is detected. Trend is a second time variable that represents the time after the interruption, and $\beta_3$ specifies the difference in the pre- and post-interruption trends of the outcome for the value of the intervention "i" at time "t."

The addition of a first-order autoregressive (p = 1) component improved model fit, while further autoregressive (p = 2) and moving average (q = 1) components reduced model fit. Therefore, a first-order autoregressive component (p = 1) was included in the final model.

We calculated the pre-interruption trend of COVID-19 case rates in prisons ($\beta_1$), the absolute change in COVID-19 case rates in prisons when community spread reached a substantial level ($\beta_2$), and the change in the trend of the COVID-19 case rates in prisons post-interruption ($\beta_3$). We report estimates of pre-trend, absolute change, and change in trend post-interruption with 95% confidence intervals (CIs).

## Sensitivity analysis

We conducted a sensitivity analysis to examine the relationship between COVID-19 community case rates and case rates within prisons among those incarcerated. To do this, we removed staff data from the prison data set (N = 35). However, it was often unclear in the data set if a test ordered from a prison was for a staff member or incarcerated individual within a prison (e.g., if the occupation field was left blank or the ordering facility did not include this information). Of the 29,605 tests ordered from a prison, 35 were staff, 28,525 were incarcerated individuals, and 1,045 were unknown (i.e., were either staff or incarcerated). We repeated all aforementioned statistical analysis using this amended data set.

## Secondary analysis

A controlled time series analysis using an autoregressive integrated moving average model was conducted to examine the changes in COVID-19 case rates in the general population among counties with and without prisons. The model can be written as:

$$\text{Outcome}_{i\ t}^{*} = \beta_0 + \beta_1{}^{*}\text{time}_t + \beta_2{}^{*}\text{PrisonCounties}_i + \beta_3{}^{*}\text{time}^{*}\text{PrisonCounties}_{i\ t}^{*} + e,$$

Where $\beta_0$ specifies the baseline COVID-19 active case rate in counties without prisons at time 1 (June 1, 2020), time is a continuous variable for counties without prisons, which has 120 time points, and $\beta_1$ specifies the trend of the outcome in counties without prisons. Prison-Counties is an indicator variable for counties with prisons, and $\beta_2$ specifies the COVID-19 active case rate in counties with prisons at time 1 (June 1, 2020) relative to counties without prison. Time*PrisonCounties is a continuous variable for counties with prisons, which has 120 time points, and $\beta_3$ specifies the trend of the outcome in counties with prisons for the value of the intervention "i" at time "t" relative to counties without prison.

**Table 1. COVID-19 testing and cases in North Carolina June 1, 2020—August 31, 2020.**

|  | Prisons | General Population |
|---|---|---|
| Population | 55,196[+] | 5,071,498 |
| COVID-19 Tests Administered | 29,605 | 621,514 |
| Positive COVID-19 Cases | 1,639 | 70,533 |
| Test Positivity | 5.5% | 11.3% |
| Mean Active Case Rate* | 427 per 100,000 | 215 per 100,000 |

[+] Incarcerated and staff.

* Sum of cases in the past 14 days divided by the incarcerated population.

## Results

Forty-six of NC's 100 counties contain prisons. From June 1, 2020—August 31, 2020, the NC general population ordered 621,514 tests and had 70,533 positive COVID-19 cases in counties with prisons (**Table 1**). Within prisons, 29,605 tests were ordered and 1,639 positive cases were found. Hence, among counties with prisons, test positivity was 11.3% for the non-incarcerated population compared to 5.5% for the incarcerated population. During this time period, the mean active COVID-19 case rate in the general population was 215 per 100,000 whereas in prisons, the mean active case rate was 427 per 100,000.

All counties with prisons reached substantial community spread (at least 50 cases per 100,000 population detected within 7 days) within this time period. The first county to reach this was Burke County on June 1, 2020 and the last county was Tyrrell County on August 25, 2020. Before substantial community spread, COVID-19 case rates increased at a rate of 5.12 cases per 100,000 (95% CI: -2.54, 12.79) (**Fig 1**). Once counties hit a level of substantial

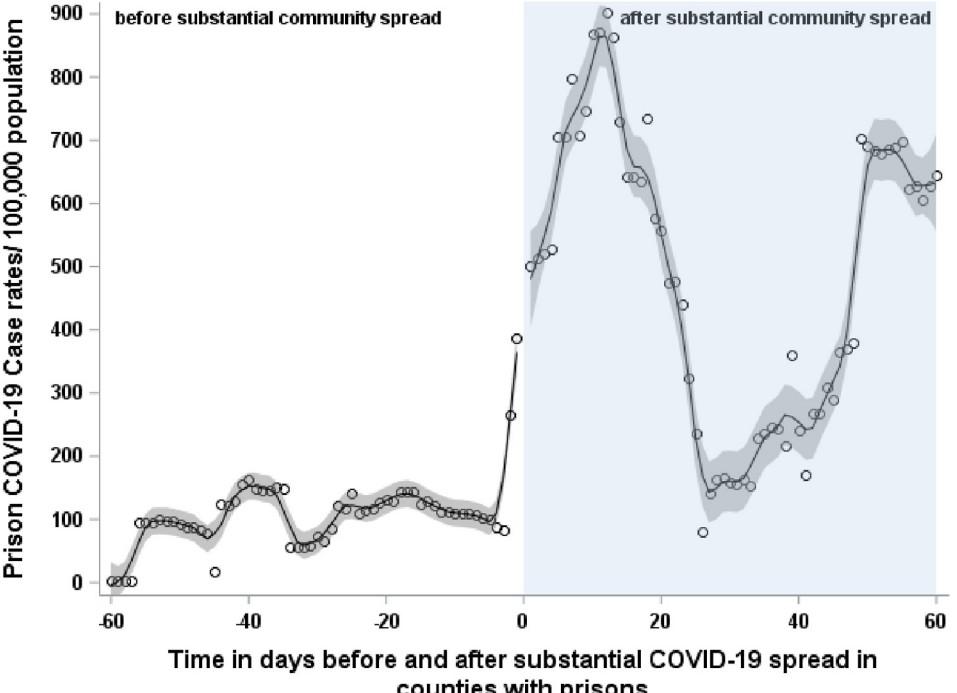

**Fig 1. Active case rates in NC prisons before and after substantial community spread in surrounding counties.**

COVID-19 spread, there was an immediate increase in the COVID-19 case rate in prisons by 118.55 cases per 100,000 (95% CI: -3.71, 240.81). This case rate declined thereafter at a rate of 2.80 cases per 100,000 (95% CI: -16.02, 10.42) over the next 60 days.

In sensitivity analyses, including only incarcerated individuals in the denominator, the results did not change substantially except that the case rates increased due to a smaller denominator.

Secondary analyses compared the general population's active COVID-19 case rate in counties with and without prisons. Between June 1 and August 31, 2020, the active COVID-19 case rate in counties without prisons increased at a rate of 2.19 (95% CI: 1.20, 3.19), and the relative change in trend in counties with prisons was 0.03 (95% CI: -1.39, 1.44). This indicates that there is no substantial difference in COVID-19 case rates in counties with and without prisons in NC.

## Discussion

In this study, we evaluated the association of substantial community COVID-19 spread on prison COVID-19 case rates in the summer of 2020 in NC. To our knowledge, this study is the first study to date to evaluate the relationship between community and prison COVID-19 spread. In the state of NC in the summer of 2020, case rates were higher in prisons than in the general population while test positivity was higher in the general population than in prisons. The lower test positivity in prisons may be due to one-time universal testing in state prisons during this time and lack of available tests in the general population, indicating that more cases were captured in prisons than the general population.

We observed that substantial community spread (50 cases per 100,000 population in the past seven days) was associated with a large immediate increase in COVID-19 case rates in prisons but that this increase was not sustained over time. This lack of an increasing trend over time is to be expected, as COVID-19 infections typically only last around two weeks. Our findings were slightly stronger when we restricted prison case rates to exclude staff cases but the overall trend remained the same. We also find that while community spread impacts case rates within prisons, prisons do not impact case rates in their surrounding communities.

Prisons have taken steps to mitigate COVID-19 including mask mandates, stopping visitations, and quarantining newly admitted individuals [4, 24]. In NC specifically, prisons suspended visitation, work release, and educational programs during the study period and provided PPE to incarcerated individuals and staff [17]. However, the majority of these steps focus on incarcerated individuals rather than staff, the primary vector of COVID-19 within facilities. These policies and practices also ignore community spread in the general population. As high community spread is likely closely tied to staff infection, and thus the infection of incarcerated individuals, focusing on community spread in the areas surrounding prisons should be included in prevention efforts. For example, community spread should be closely monitored and more stringent measures relevant to staff should be taken as community spread increases with new variants in order to prevent COVID-19 outbreaks within facilities.

As COVID-19 vaccination efforts have increased to prevent harmful outcomes (e.g., hospitalizations, death), especially with increased threats from new variants that cause breakthrough infections, understanding the relationship between community and prison COVID-19 spread is even more important. For example, NC state prisons began offering the COVID-19 vaccination to incarcerated individuals with 19,722 of 28,405 (69%) individuals having received at least one dose of the by September 10, 2021. However, in comparison, 7,291 of 16,100 (45%) staff members had received at least one dose of the vaccine by September 10, 2021 [1]. There are many reports of staff refusing COVID-19 vaccinations, which has direct implications for

the health of incarcerated individuals [25]. Beyond staff and incarcerated individuals' vaccination status, it is important to consider the vaccination status of surrounding counties. For example, by September 10, 2021, in Wake County, which contains three prisons, 66% of the population had received at least one dose, but in Tyrrell County, which contains one prison, only 47% had received at least one dose [26].

Beyond incremental and stringent COVID-19 mitigation measures, it is important that institutions focus on decarceration. For example, there is a need for compassionate releases, with 11% of the prison population being above age 55 and many suffering from severe chronic conditions, both of which increase their risk of severe COVID-19 [27]. More broadly, policy efforts should be aimed at decarceration to reduce the number of those sentenced to the carceral system and increase investments in communities suffering from mass incarceration [28].

Our study has limitations that must be considered in interpreting the impact of community case rates on prison case rates. First, our assessment of COVID-19 tests related to prisons is imperfect. Tests were considered to be associated with a prison if the ordering facility was the name or address of a state, private, or federal prison in NC. It is possible that staff received COVID-19 tests outside of the prison and it is not possible to link these tests with the prison they were employed at. While a lawsuit passed to mandate testing during Summer of 2020, multiple counties did not report many tests during this time, indicating that there is likely asymptomatic spread of COVID-19 that this study does not capture. Second, the type of test performed was not available. Additionally, general populations in counties also likely have asymptomatic spread. This indicates that our results are likely a conservative estimate of the relationship between community and prison transmission.

## Conclusions

COVID-19 continues to devastate both prisons and communities. This study presents the first state-wide evidence that community COVID-19 spread contributes to COVID-19 case rates within prisons. The public health community must recognize that prisons are not separate from communities and that community health impacts carceral settings.

## Author Contributions

**Conceptualization:** Katherine LeMasters, Shabbar Ranapurwala, Morgan Maner, Kathryn M. Nowotny, Meghan Peterson, Lauren Brinkley-Rubinstein.

**Data curation:** Katherine LeMasters, Lauren Brinkley-Rubinstein.

**Formal analysis:** Katherine LeMasters, Shabbar Ranapurwala.

**Funding acquisition:** Kathryn M. Nowotny, Lauren Brinkley-Rubinstein.

**Methodology:** Katherine LeMasters, Shabbar Ranapurwala.

**Resources:** Lauren Brinkley-Rubinstein.

**Supervision:** Shabbar Ranapurwala.

**Writing – original draft:** Katherine LeMasters, Shabbar Ranapurwala, Morgan Maner, Kathryn M. Nowotny, Meghan Peterson, Lauren Brinkley-Rubinstein.

**Writing – review & editing:** Katherine LeMasters, Shabbar Ranapurwala, Morgan Maner, Kathryn M. Nowotny, Meghan Peterson, Lauren Brinkley-Rubinstein.

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
