## [Decision Letter · Decision Letter 0]

19 Dec 2021

PONE-D-21-34261COVID-19 Community Spread and Consequences for Prison Case RatesPLOS ONE

Dear Dr. LeMasters,

Thank you for submitting your manuscript to PLOS ONE. After careful consideration, we feel that it has merit but does not fully meet PLOS ONE’s publication criteria as it currently stands. Therefore, we invite you to submit a revised version of the manuscript that addresses the points raised during the review process; comments from both reviewers are detailed below.

Please submit your revised manuscript by Feb 02 2022 11:59PM. I appreciate that this decision is being sent to you just before the Christmas and New Year break. If you will need more time than this to complete your revisions, please reply to this message or contact the journal office at plosone@plos.org. Please include the following items when submitting your revised manuscript:A rebuttal letter that responds to each point raised by the academic editor and reviewer(s). You should upload this letter as a separate file labeled 'Response to Reviewers'.A marked-up copy of your manuscript that highlights changes made to the original version. You should upload this as a separate file labeled 'Revised Manuscript with Track Changes'.An unmarked version of your revised paper without tracked changes. You should upload this as a separate file labeled 'Manuscript'.

We look forward to receiving your revised manuscript.

Kind regards,

Steph Scott

Academic Editor

PLOS ONE

Journal Requirements:

Reviewers' comments:

Reviewer's Responses to Questions

**Comments to the Author**

1. Is the manuscript technically sound, and do the data support the conclusions?

Reviewer #1: Partly

Reviewer #2: Yes

2. Has the statistical analysis been performed appropriately and rigorously? 

Reviewer #1: Yes

Reviewer #2: I Don't Know

3. Have the authors made all data underlying the findings in their manuscript fully available?

Reviewer #1: Yes

Reviewer #2: No

4. Is the manuscript presented in an intelligible fashion and written in standard English?

Reviewer #1: Yes

Reviewer #2: Yes

5. Review Comments to the Author

Reviewer #1: The work deals with the study of the relationships between the spread of covid-19 in prisons and the relationships with the spread in the population of the same geographical area. The work is well structured, albeit limited to one state in the United States of America.

Some changes are needed.

1) clarify the difference between jail a prison (in everyday language they are synonyms).

2) Regarding the construction and conduct of the study, indicate explicitly which COVID-19 tests were used. Clarify which type of test was performed on the two populations (molecular PCR tests, rapid tests, both, et). If they have been used in different cases and in different populations, indicate this explicitly and exclude cases that do not correspond to the modality considered "included" in the method. Do not compare results from different tests.

Reviewer #2: This paper examines the important topic of the spread of COVID-19 in prisons where local COVID-19 cases increased. While I agree the topic is important, overall, the paper has some writing flaws and is lacking a well-reasoned rationale, which limited my enthusiasm for the manuscript.

Abstract

No suggestions for the abstract.

Introduction

Global: The paper is not well reasoned, and the rationale is not well formed for the paper. For example, the paragraph on page 3 which starts “Early in the COVID-19 pandemic,” provides information on jails, Federal Bureau of prisons, and single site outbreaks with nothing tying any of the information. Please tighten up the introduction with a strong rationale for THIS study.

• Page 3, par1; The word most should be changed to many: “Most single-site cluster outbreaks”

• Page 3, par 1; The word “are” should be were: “As of September 10, 2021 over 421,000 people who are incarcerated”

• Page 3, par 1; This last part of paragraph one is confusing “Risk is heightened due to a confluence of several factors. Prisons detain people who have a higher burden of chronic disease (3). In addition, the built environment of prison facilities, where people often live in close, overcrowded facilities, make common prevention strategies such as social distancing nearly impossible.” Are the authors trying to provide an example of the “several factors”? At a minimum it needs to be reworded but more importantly it needs to convey the problem you are trying to solve with this study.

• Page 3, par 2; The mention of systemic racism is (I believe) followed by an example showing how systemic racism operates. This needs to be made more clear.

• Another overall note that the rationale for the study is very weak and needs to be improved. The introduction is all over the place with no real focus on what this paper attempts to show. It feels as though the introduction is an afterthought to the rest of the paper.

Method

• Page 5, par 2; Please explain why you chose to use the denominators from Vera and NC website. Is this common procedure used in other studies?

• It may be helpful to have the number of COVID deaths during your study period if the data is available to the research team.

• It may be helpful to cite some other studies that have used your analytic technique so that the reader can become comfortable with the methodology.

Results

• The analytic methodology is confusing and perhaps you could make more clear in your explanation. In the example of Burke County and the “substantial community spread” identifier you state that it reached that rate on the first day of data collection. You earlier state that you restricted data collection to “Between June 1 and Aug 31”, if that is the case then how do you have the data for the 60 days prior to substantial spread?

• The rate of only 5.5% of people in prison contracting covid seems extremely low. Where these individuals on lockdown? Was PPE available?

Discussion

• Page 1, par 2; You don’t have evidence in this study to make this claim “The lower

test positivity in prisons is likely due to one-time universal testing in state prisons during this

time and lack of available tests in the general population, indicating that more cases were

captured in prisons than the general population.” Suggest change the word from is likely to may be.

• Please elaborate more on this finding: In the state of NC in the summer of 2020, case rates were higher in prisons than in the general population while test positivity was higher in the general population than in prisons.

• Page 10, par 2; please reword this sentence: “As COVID-19 vaccination efforts have increased, the relationship between community and prison COVID-19 spread is even more important and preventable.” It is unclear why this makes it more important or preventable?

• Suggest changing the wording of “should be of utmost importance” with “should be included in the prevention efforts”.

• I would remove this sentence. “For example, there is a need to end pretrial detention and cash bail, as it creates a large amount of churn in the jail population and significantly contributes to COVID-19 jail and community spread.” While this may be true, your study focused on prison not jails.

Tables and Figures

• Nothing to note.

6. PLOS authors have the option to publish the peer review history of their article (what does this mean?). If published, this will include your full peer review and any attached files.

Reviewer #1: No

Reviewer #2: **Yes: **Noel A Vest

---

## [Author Response · Author response to Decision Letter 0]

7 Feb 2022

Reviewer #1: The work deals with the study of the relationships between the spread of covid-19 in prisons and the relationships with the spread in the population of the same geographical area. The work is well structured, albeit limited to one state in the United States of America.

Some changes are needed.

1) clarify the difference between jail a prison (in everyday language they are synonyms).

Thank you for this comment, we have added this when we state that prisons have more stable populations than jails (introduction, third paragraph, third sentence). 

2) Regarding the construction and conduct of the study, indicate explicitly which COVID-19 tests were used. Clarify which type of test was performed on the two populations (molecular PCR tests, rapid tests, both, et). If they have been used in different cases and in different populations, indicate this explicitly and exclude cases that do not correspond to the modality considered "included" in the method. Do not compare results from different tests.

Thank you for this suggestion. We agree that we should not be comparing results from different tests, but these data are not available for all tests. We have added this into the limitations section (discussion, last paragraph, sixth sentence).

Reviewer #2: This paper examines the important topic of the spread of COVID-19 in prisons where local COVID-19 cases increased. While I agree the topic is important, overall, the paper has some writing flaws and is lacking a well-reasoned rationale, which limited my enthusiasm for the manuscript.

Abstract

No suggestions for the abstract.

Introduction

Global: The paper is not well reasoned, and the rationale is not well formed for the paper. For example, the paragraph on page 3 which starts “Early in the COVID-19 pandemic,” provides information on jails, Federal Bureau of prisons, and single site outbreaks with nothing tying any of the information. Please tighten up the introduction with a strong rationale for THIS study.

Thanks for the suggestion. We have now revised this paragraph and the next one, to tie the information and strengthen the rationale for our study (introduction, third and fourth paragraph). Specifically, we had included the example of the study in the Chicago jail because it is the only study to-date that explores the relationship between carceral facility case rates and surrounding community case rates. However, we recognize that this is largely due to jail churn. Because prisons have less population turnover, they are thought to be more separate from community spread of COVID-19. However, prison populations are frequently exposed to staff that live in the surrounding community, so we thought it was critically important to understand the relationship between case rates in prisons and community spread. Focusing on staff cases alone is insufficient, as many cases go undetected. Focusing on the surrounding community case rates provide a fuller picture of how community case rates impact prisons. We have expanded on this and hope that this strengthens our rationale. Please let us know if anything remains unclear. 

• Page 3, par1; The word most should be changed to many: “Most single-site cluster outbreaks”

Thank you for catching this, we have changed it to ‘many.’

• Page 3, par 1; The word “are” should be were: “As of September 10, 2021 over 421,000 people who are incarcerated”

Thank you for catching this, we have changed it to ‘were’ and made the rest of the sentence past tense.

• Page 3, par 1; This last part of paragraph one is confusing “Risk is heightened due to a confluence of several factors. Prisons detain people who have a higher burden of chronic disease (3). In addition, the built environment of prison facilities, where people often live in close, overcrowded facilities, make common prevention strategies such as social distancing nearly impossible.” Are the authors trying to provide an example of the “several factors”? At a minimum it needs to be reworded but more importantly it needs to convey the problem you are trying to solve with this study.

Thank you for this comment. Yes, we are providing examples here and have changed the sentence construction to help clarify this. We are simply providing motivation here as to why COVID risk is heightened in prisons here. We have clarified the contribution of the study (i.e., the problem we are trying to solve) further down in paragraph three of the introduction and hope that the first paragraph simply provides context as to why COVID risk is heightened in prisons.

• Page 3, par 2; The mention of systemic racism is (I believe) followed by an example showing how systemic racism operates. This needs to be made more clear.

Thank you for this comment. We have reworded these sentences.

• Another overall note that the rationale for the study is very weak and needs to be improved. The introduction is all over the place with no real focus on what this paper attempts to show. It feels as though the introduction is an afterthought to the rest of the paper.

Thank you again for your suggestions. As noted above under your Global Comment on the introduction, we have now revised the introduction to clarify the rationale (introduction, third and fourth paragraph). Please let us know if there is more that we should address. 

Method

• Page 5, par 2; Please explain why you chose to use the denominators from Vera and NC website. Is this common procedure used in other studies?

Thank you for this question. Yes, Vera collects the most updated data on prison population numbers and it is used in many analyses. Please see the following citations as examples (https://doi.org/10.1186/s40352-020-00125-3, https://doi.org/10.1186/s12889-021-11077-0). However, staff population numbers are not provided publicly, so we received a report from the North Carolina Department of Public Safety of staff counts by institution. 

• It may be helpful to have the number of COVID deaths during your study period if the data is available to the research team.

Thank you for this suggestion - we agree that it would be helpful to document COVID deaths during this period and to compare case fatality rates between the incarcerated and general population. However, this data is not available to us, unfortunately. While we are able to find death data for the incarcerated population and general population, it is not in our analytic data set from NC DHHS. This poses a few problems. First, our analytic data set combines staff and incarcerated data and then separates them in sensitivity analyses. However, staff death data from COVID-19 are not reported from NC’s Department of Corrections. So, it is not possible for us to understand staff deaths. Second, because we do not have deaths in the same file as tests and positive cases, we are unable to know if deaths occurring from 6/1/20-8/31/20 are due to positive cases detected during this period or beforehand. Given that our analysis is restricted to this time period, it would be important that deaths would be documented from this time period as well. That said, the Covid Prison Project provides data that 26 deaths occurred among incarcerated individuals in NC from 6/1/20-8/31/20, resulting in a mortality rate of 82.5 per 100,000. However, given that we cannot link this to cases detected during the study period, we cannot present a case fatality rate.

• It may be helpful to cite some other studies that have used your analytic technique so that the reader can become comfortable with the methodology.

Thank you for this suggestion. In addition to Bernal et. al., 2017 - which is a methodological paper, we have added two citations that are substantive papers using interrupted time series. We have also added the following sentence: “This is an appropriate and useful method when assessing changes due to an intervention or events that occur at a clearly defined point in time (e.g., the date the community reached substantial community spread).”

Results

• The analytic methodology is confusing and perhaps you could make more clear in your explanation. In the example of Burke County and the “substantial community spread” identifier you state that it reached that rate on the first day of data collection. You earlier state that you restricted data collection to “Between June 1 and Aug 31”, if that is the case then how do you have the data for the 60 days prior to substantial spread?

Thank you for catching this. As we state in the paper, we had restricted the data to the summer months due to more accurate testing. We then used up to the 60 days prior to and after reaching substantial spread but restricted this to the summer months as well. We have clarified this in our methods section. 

• The rate of only 5.5% of people in prison contracting covid seems extremely low. Where these individuals on lockdown? Was PPE available?

Thank you for this. It was not 5.5% that contracted COVID - it was 5.5% of tests conducted in prisons being positive (i.e., test positivity). There were 1,639 positive tests out of 55,996 individuals - resulting in a mean active case rate that is double that of the general population. However, as we state, many cases likely were not captured as testing remained insufficient during this time. 

Regarding lockdown, there were no facility-wide quarantines during that time but the Department of Correction stated that they limited the movement of incarcerated people by suspending visitation, work release, and educational programs from April through August 2020. Regarding PPE, the Department of Public Safety website states that face masks were first distributed to incarcerated people and staff in March of 2020. Surgical masks, N95 masks, face shields, and gowns were provided for staff. Face masks were considered mandatory for staff starting in April of 2020 (https://www.ncdps.gov/our-organization/adult-correction/adult-correction-actions-covid-19#may--20). However, we do not have information about the enforcement of PPE use. We have added a sentence to the discussion paragraph 3 that these steps were taken in NC during the study period. 

Discussion

• Page 1, par 2; You don’t have evidence in this study to make this claim “The lower

test positivity in prisons is likely due to one-time universal testing in state prisons during this

time and lack of available tests in the general population, indicating that more cases were

captured in prisons than the general population.” Suggest change the word from is likely to may be.

We have softened the language here to ‘may be.’

• Please elaborate more on this finding: In the state of NC in the summer of 2020, case rates were higher in prisons than in the general population while test positivity was higher in the general population than in prisons.

In this statement, we stated one of the main findings of our study presented in the first paragraph of the results. Following this statement, we discussed possible explanations for this finding in the next sentence, which you have referred to in your prior comment.

• Page 10, par 2; please reword this sentence: “As COVID-19 vaccination efforts have increased, the relationship between community and prison COVID-19 spread is even more important and preventable.” It is unclear why this makes it more important or preventable?

We have revised the statement to reflect that it is important to understand that lack of vaccination among the prison staff and the surrounding communities can lead to greater burden of COVID-19 among the incarcerated population (discussion, paragraph 4).

• Suggest changing the wording of “should be of utmost importance” with “should be included in the prevention efforts”.

We have made this edit.

• I would remove this sentence. “For example, there is a need to end pretrial detention and cash bail, as it creates a large amount of churn in the jail population and significantly contributes to COVID-19 jail and community spread.” While this may be true, your study focused on prison not jails.

We have deleted this sentence. 

Tables and Figures

• Nothing to note.

---

## [Decision Letter · Decision Letter 1]

28 Mar 2022

COVID-19 Community Spread and Consequences for Prison Case Rates

PONE-D-21-34261R1

Dear Dr. LeMasters,

We’re pleased to inform you that your manuscript has been judged scientifically suitable for publication and will be formally accepted for publication once it meets all outstanding technical requirements.

Kind regards,

Steph Scott

Academic Editor

PLOS ONE

Additional Editor Comments (optional):

Reviewers' comments:

Reviewer's Responses to Questions

**Comments to the Author**

1. If the authors have adequately addressed your comments raised in a previous round of review and you feel that this manuscript is now acceptable for publication, you may indicate that here to bypass the “Comments to the Author” section, enter your conflict of interest statement in the “Confidential to Editor” section, and submit your "Accept" recommendation.

Reviewer #1: (No Response)

Reviewer #2: All comments have been addressed

2. Is the manuscript technically sound, and do the data support the conclusions?

Reviewer #1: (No Response)

Reviewer #2: Yes

3. Has the statistical analysis been performed appropriately and rigorously? 

Reviewer #1: (No Response)

Reviewer #2: I Don't Know

4. Have the authors made all data underlying the findings in their manuscript fully available?

Reviewer #1: (No Response)

Reviewer #2: No

5. Is the manuscript presented in an intelligible fashion and written in standard English?

Reviewer #1: (No Response)

Reviewer #2: Yes

6. Review Comments to the Author

Reviewer #1: (No Response)

Reviewer #2: The authors have made adequate changes to the paper and I have no further changes or suggestions to note.

7. PLOS authors have the option to publish the peer review history of their article (what does this mean?). If published, this will include your full peer review and any attached files.

Reviewer #1: No

Reviewer #2: **Yes: **Noel A Vest

---

## [Editor Report · Acceptance letter]

4 Apr 2022

PONE-D-21-34261R1 

COVID-19 community spread and consequences for prison case rates 

Dear Dr. LeMasters:

I'm pleased to inform you that your manuscript has been deemed suitable for publication in PLOS ONE. Congratulations! Your manuscript is now with our production department. 

Kind regards, 

on behalf of

Dr. Steph Scott 

Academic Editor

PLOS ONE